DOI: 10.1038/s41467-018-04609-7　　**OPEN**

# Targeting repair pathways with small molecules increases precise genome editing in pluripotent stem cells

Stephan Riesenberg[1] & Tomislav Maricic[1]

A now frequently used method to edit mammalian genomes uses the nucleases CRISPR/Cas9 and CRISPR/Cpf1 or the nickase CRISPR/Cas9n to introduce double-strand breaks which are then repaired by homology-directed repair using DNA donor molecules carrying desired mutations. Using a mixture of small molecules, the "CRISPY" mix, we achieve a 2.8- to 7.2-fold increase in precise genome editing with Cas9n, resulting in the introduction of the intended nucleotide substitutions in almost 50% of chromosomes or of gene encoding a blue fluorescent protein in 27% of cells, to our knowledge the highest editing efficiency in human induced pluripotent stem cells described to date. Furthermore, the CRISPY mix improves precise genome editing with Cpf1 2.3- to 4.0-fold, allowing almost 20% of chromosomes to be edited. The components of the CRISPY mix do not always increase the editing efficiency in the immortalized or primary cell lines tested, suggesting that employed repair pathways are cell-type specific.

[1] Department of Evolutionary Genetics, Max-Planck-Institute for Evolutionary Anthropology, Deutscher Pl. 6, 04103 Leipzig, Germany. Correspondence and requests for materials should be addressed to S.R. (email: stephan.riesenberg@eva.mpg.de)

Embryonic stem cells (ESCs) and induced pluripotent stem cells (iPSCs) have the potential to differentiate into many types of adult cells and have become an important tool, e.g., for disease modeling, drug development, and tissue repair[1,2]. Stem cells are especially powerful in combination with the ability to precisely and efficiently edit DNA with the CRISPR technology. Often, multiple edits are required to test sets of variant alleles (e.g., epistatic interaction that may be associated with a certain disease), but this requires development of methods that increase the editing efficiency of stem cells.

The bacterial nuclease CRISPR/Cas9 is now frequently used to accurately cut chromosomal DNA sequences in eukaryotic cells. The resulting DNA double-strand breaks (DSBs) are repaired by two competing pathways: non-homologous end joining (NHEJ) and homology-directed repair (HDR) (Fig. 1). In NHEJ, the first proteins to bind the cut DNA ends are Ku70/Ku80, followed by DNA protein kinase catalytic subunit (DNA-PKcs)[3]. The kinase phosphorylates itself and other downstream effectors at the repair site, which results in joining of the DNA ends by DNA ligase IV[4]. If this canonical NHEJ is repressed, the alternative NHEJ pathway becomes active[5], which, among other proteins, requires Werner syndrome ATP-dependent helicase. HDR is initiated when the MRN complex binds to the DSB[3]. In this case, DNA endonuclease RBBP8 (CtIP) removes nucleotides at the 5′-ends. Further resection produces long 3′ single-stranded DNA (ssDNA) overhangs on both sides of the DNA break[4]. These are coated and stabilized by the replication protein A (RPA) complex, followed by generation of a RAD51 nucleoprotein filament[3]. RAD52 facilitates replacement of RPA bound to ssDNA with RAD51 and promotes annealing to a homologous donor DNA[6]. Subsequent DNA synthesis results in precisely repaired DNA. In HDR, the protein kinase ataxia telangiectasia mutated (ATM) has a major role in that it phosphorylates at least 12 proteins involved in the pathway[3].

As NHEJ of Cas9-induced DSBs is error prone and frequently introduces short insertions and deletions (indels) at the cut site, it is useful for knocking out a targeted gene. In contrast, HDR allows precise repair of a DSB by using a homologous donor DNA. If the donor DNA provided in the experiment carries mutations, these will be introduced into the genome (precise genome editing). Repair with homologous ssDNA or double-stranded DNA (dsDNA) has been suggested to engage different pathways[7]. We will refer to targeted nucleotide substitutions using ssDNA donors (ssODNs) as "TNS" and targeted insertion of cassettes using dsDNA donors as knock-ins, respectively. In order to introduce a DSB, Cas9 requires the nucleotide sequence NGG (a "PAM" site) in the target DNA. Targeting of Cas9 is further determined by a guide RNA (gRNA) complementary to 20 nucleotides adjacent to the PAM site. However, the Cas9 may also cut the genome at sites that carry sequence similarity to the gRNA[8]. One strategy to reduce such off-target cuts is to use a mutated Cas9 that introduces single-stranded nicks instead of DSBs (Cas9n)[9]. Using two gRNAs to introduce two nicks on opposite DNA strands in close proximity to each other (double nicking) will result in a staggered DSB at the desired location, while reducing the risk of off-target DSBs, because two nicks close enough to cause a DSB are unlikely to occur elsewhere in the genome. Another strategy is to use Cpf1[10], a nuclease that introduces staggered cuts near T-rich PAM sites and causes less off-target DSBs than Cas9[11,12].

Efficiencies of TNS in human stem cells range from 15% down to as low as 0.5%[13,14] making the isolation of edited homozygous clones challenging. Several studies have tried to increase precise genome-editing efficiency by promoting HDR or decreasing NHEJ. Synchronization of cells to the S or $G_2$/M phase when homologous recombination occurs increases TNS efficiency in HEK cells (from 26% to 38%), human primary neonatal fibroblast (undetectable to 0.6%), and human ESCs (hESCs) (undetectable to 1.6%)[15], and knock-in efficiency in hESCs (from 7% to 41% after sorting)[16]. Improved knock-in efficiency was also achieved in HEK cells by suppressing repair proteins like Ku70/80 and DNA ligase IV with small interfering RNA (from 5% to 25%) or

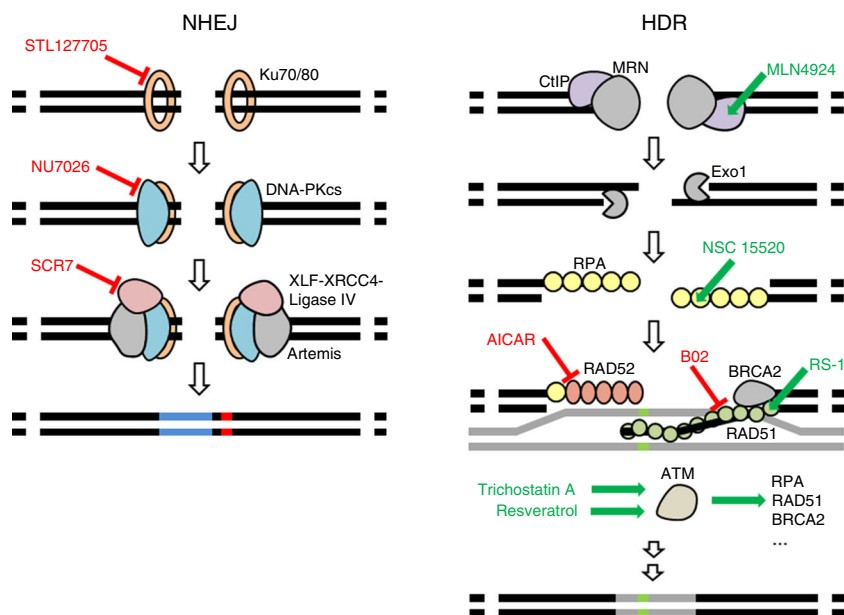

**Fig. 1** Small molecules described or anticipated to target key proteins of NHEJ and HDR. Proteins are labeled with black text and inhibitors and enhancing small molecules are marked red and green, respectively. STL127705, NU7026, or SCR7 have been described to inhibit Ku70/80, DNA-PK, or DNA ligase IV, respectively. MLN4924, RS-1, Trichostatin A, or Resveratrol have been described to enhance CtIP, RAD51, or ATM, respectively. NSC 15520 has been described to block the association of RPA to p53 and RAD9. AICAR is an inhibitor of RAD52 and B02 is an ihibitor of RAD51. For simplicity, some proteins and protein interactions are not depicted

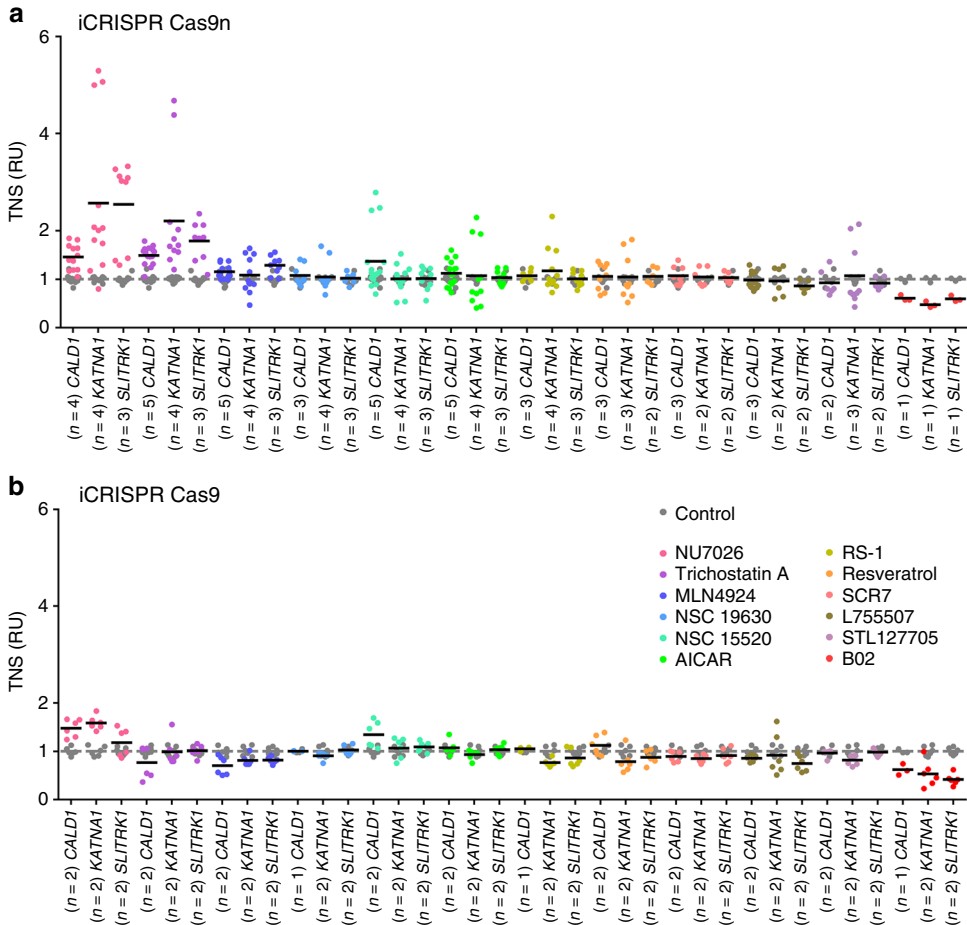

**Fig. 2** Effects of small molecules on targeted nucleotide substitution (TNS) efficiency in iCRISPR hiPSCs. Shown are TNS efficiencies in *CALD1*, *KATNA1* and *SLITRK1* with Cas9n (**a**) and Cas9 (**b**) in 409-B2 iCRISPR hiPSCs. TNS efficiency is given in relative units (RU) with the mean of controls set to 1 to account for varying efficiency in different loci. Shown are technical replicates of *n* independent experiments. Data from Fig. 3 and Supplementary Fig. 3 are included. Gray and black bars represent the mean of the control and the respective small molecule, respectively. Concentrations used were 20 μM NU7026, 0.01 μM Trichostatin A, 0.5 μM MLN4924, 1 μM NSC 19630, 5 μM NSC 15520, 20 μM AICAR, 1 μM RS-1, 1 μM Resveratrol, 1 μM SCR7, 5 μM L755507, 5 μM STL127685, and 20 μM B02. Mean absolute percentages of TNS and indels of all technical replicates are shown in Supplementary Table 4

by coexpression of adenovirus type 5 proteins 4E1B55K and E4orf6, which mediate degradation of DNA ligase IV among other targets (from 5% to 36%)[17].

Several small molecules have been used to increase precise genome editing in various cell lines[16–26] (Supplementary Table 1). In summary, inhibitors of DNA-PK (NU7026 and NU7441) tend to increase precise genome-editing efficiency in different cell lines, while the effects of SCR7, L755507, and RS-1 are not consistent between cell lines. In this study, we systematically screen several small molecules and find a small-molecule mix that additively increases TNS and gene fragment insertion efficiency in pluripotent stem cells, when a DSB with 5′-overhangs is introduced with Cas9n double nicking or Cpf1 and a donor DNA is provided as ssODN. We also find that small molecules can have non-identical and even opposite effects on precise genome-editing efficiencies in different cell types, possibly explaining the inconsistencies reported in the literature.

## Results
**Individual small-molecule effects on editing efficiency.** Here we test the above as well as other small molecules with respect to

their efficiency to induce TNS in human iPSCs (hiPSCs). We identified additional molecules interacting with repair proteins listed in the REPAIRtoire database[27] by literature and database (ChEMBL[28]) search. The additional molecules we test, which have been described to either block NHEJ/alternative NHEJ or to activate or increase the abundance of proteins involved in HDR/damage-dependent signaling (Fig. 1 and Supplementary Table 2), are as follows: NU7026, Trichostatin A, MLN4924, NSC 19630, NSC 15520, AICAR, Resveratrol, STL127685, and B02.

We tested these molecules in hiPSC lines that we generated that carry doxycycline-inducible Cas9 (iCRISPR-Cas9) and Cas9 nickase with the D10A mutation (iCRISPR-Cas9n) integrated in their genomes[13]. After the delivery of gRNA (duplex of chemically synthesized crRNA and tracrRNA) and ssODN, cells were treated with small molecules for 24 h, expanded, their DNA was collected, targeted loci sequenced, and editing efficiency quantified (Supplementary Fig. 1 and Supplementary Fig. 2).

We tested the effect of different concentrations of each molecule on TNS in the three genes *CALD1*, *KATNA1*, and *SLITRK1* in 409B2 iCRISPR-Cas9n hiPSCs. For further experiments we used the concentration that gave the highest frequency of TNS, or if two or more concentrations gave a similarly high frequency we chose the lowest concentration (Supplementary

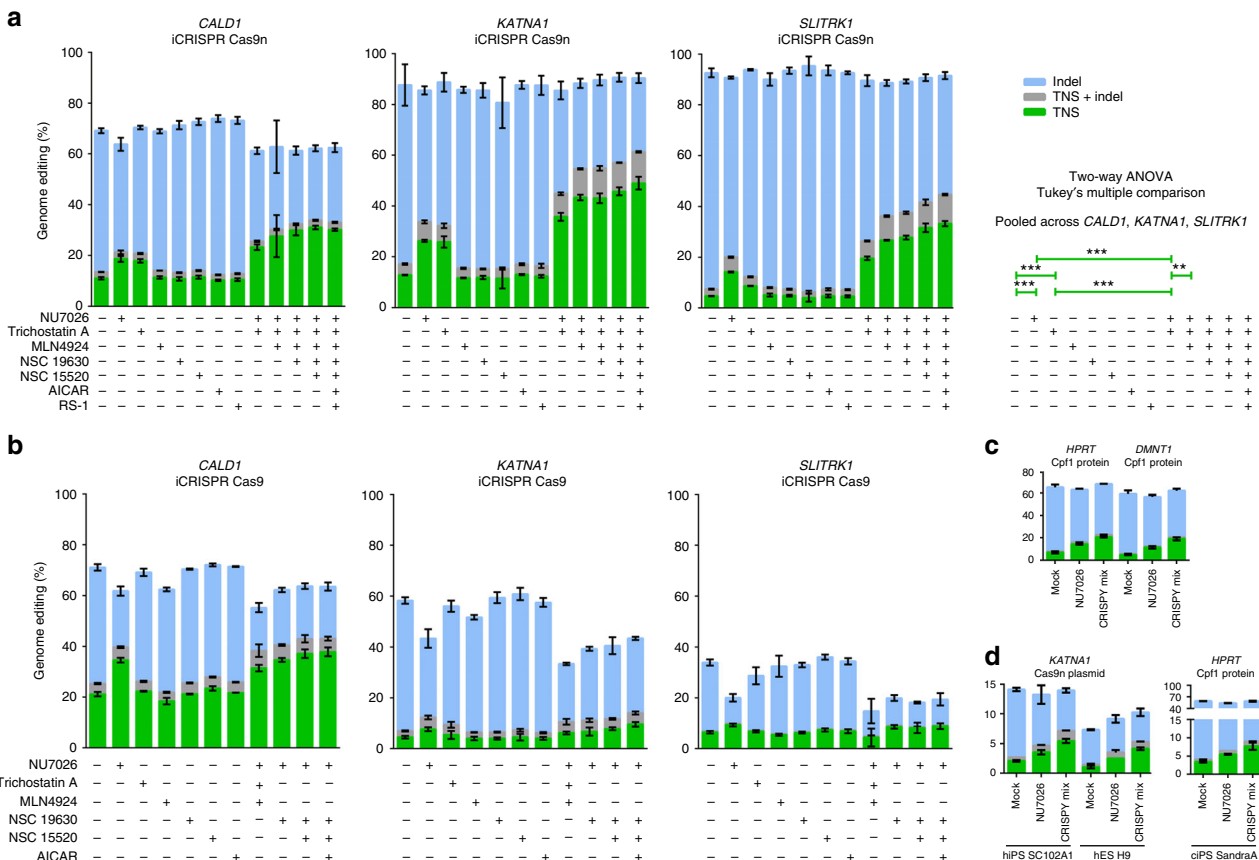

**Fig. 3** Impact of small-molecule combinations on targeted nucleotide substitution (TNS) efficiency in iPSCs and hESCs. Shown are TNS efficiencies in *CALD1*, *KATNA1*, and *SLITRK1* with Cas9n and Cas9, and in *HPRT* and *DNMT1* with Cpf1. Small molecules have an additive effect on TNS efficiency with Cas9n (**a**) but not with Cas9 (**b**) in the 409-B2 iCRISPR hiPSC lines. TNS of *HPRT* and *DNMT1* in 409-B2 hiPSCs with recombinant Cpf1 was increased using the CRISPY mix as well (**c**). Using the CRISPY mix, TNS efficiency was also increased in SC102 A1 hiPSCs and H9 hESCs with plasmid-delivered Cas9n-2A-GFP (GFP-FACS enriched), and in chimpanzee SandraA ciPSCs with recombinant Cpf1 (**d**). Shown are TNS, TNS + indels, and indels with green, gray, or blue bars, respectively. Error bars show the SD of three technical replicates for **a**, **b**, and **c**, and two technical replicates for **d**. Concentrations used were 20 μM of NU7026, 0.01 μM of Trichostatin A, 0.5 μM MLN4924, 1 μM NSC 19630, 5 μM NSC 15520, 20 μM AICAR, and 1 μM RS-1. CRISPY mix indicates a small-molecule mix of NU7026, Trichostatin A, MLN4924, and NSC 15520. Statistical significances of TNS efficiency changes was determined using a two-way ANOVA and Tukey's multiple comparison pooled across the three genes *CALD1*, *KATNA1*, and *SLITRK1*. Genes and treatments were treated as random and fixed effect, respectively. *P*-values are adjusted for multiple comparison (**$P \leq 0.01$, ***$P \leq 0.001$). Overall, there was a clear treatment effect (F(12, 24) = 32.954, $P \leq 0.001$)

Fig. 3). Dependent on the targeted gene, we found that NU7026 increased TNS 1.5- to 2.5-fold in Cas9n cells (Fig. 2a and Supplementary Table 4) and 1.2- to 1.6-fold in Cas9 cells (Fig. 2b and Supplementary Table 4). Trichostatin A increased TNS 1.5- to 2.2-fold in Cas9n cells, whereas no increase was seen in Cas9 cells. MLN4924 increased TNS 1.1- to 1.3-fold in Cas9n cells, whereas it slightly reduced TNS in Cas9 cells. NSC 15520 increased TNS of *CALD1* 1.4-fold and 1.3-fold in Cas9n and Cas9 cells, respectively, but had no effect on TNS of *KATNA1* and *SLITRK1*. NSC 19630, AICAR, RS-1, Resveratrol, SCR7, L755507, and STL127685 showed no clear effect on TNS frequency in the three genes in Cas9n cells and had no effect or decreased TNS in Cas9 cells. B02 reduced TNS in all three genes in both cell lines (Fig. 2).

**Additive effect of small molecules**. To test whether combinations of these compounds enhance TNS, we combined compounds that individually increased TNS for at least one gene in Cas9n cells and never decreased TNS. Those are NU7026,

Trichostatin A, MLN4924, NSC 19630, NSC 15520, AICAR, and RS-1. The results are shown in Fig. 3a, b. Treatment with NU7026 or Trichostatin A resulted in 2.3- or 1.8-fold higher TNS in Cas9n cells (Tukey's pair-wise post-hoc comparisons: $p < 0.001$) (Fig. 3a) and combinations of NU7026 and Trichostatin A resulted in 1.3 to 1.6 times higher TNS than with either compound alone ($p < 0.001$). Addition of MLN4924 to the mix of NU7026 and Trichostatin A lead to an additional 1.3-fold increase in TNS ($p < 0.01$). Further addition of NSC 15520 slightly increased the mean TNS in Cas9n cells, without reaching statistical significance. Addition of NSC 19630, AICAR, and RS-1 had no measurable effect on TNS. We conclude that the mix of small molecules that increases the frequency of TNS with Cas9n the most (although we admittedly could not test all combinatorial possibilities) is a combination of NU7026 (20 μM), Trichostatin A (0.01 μM), MLN4924 (0.5 μM), and NSC 15520 (5 μM). This "CRISPY" nickase mix results in an increase of TNS of 2.8-fold (from 11% to 31%) for *CALD1*, 3.6-fold (from 12.8% to 45.8%) for *KATNA1*, and 6.7-fold (from 4.7% to 31.6%) for *SLITRK1* in the iCRISPR 409-B2 iPSC line (NSC 19630 has no effect on TNS efficiency).

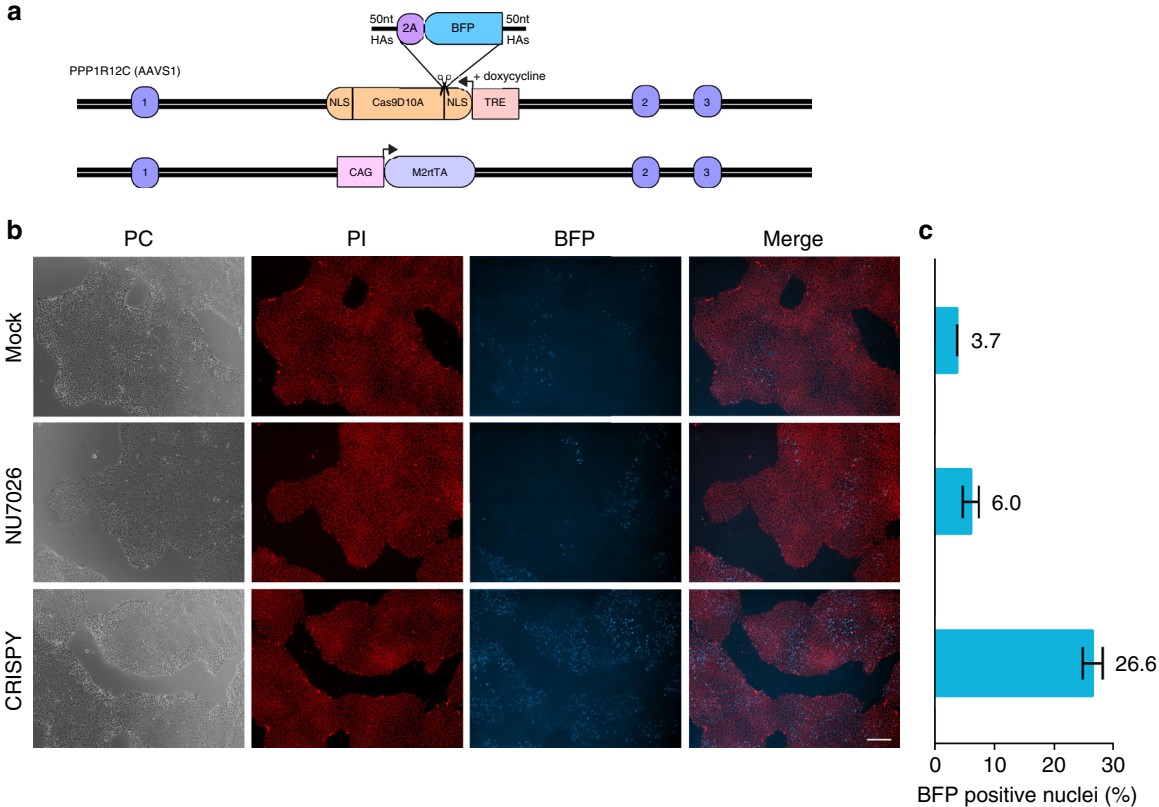

**Fig. 4** Impact of the CRISPY mix on gene fragment insertion efficiency in iCRISPR hiPSCs. Shown is the insertion efficiency of a gene fragment coding for a blue fluorescent protein (BFP) in the heterozygous AAVS1 iCRISPR locus using a single-stranded DNA donor in 409-B2 iCRISPR-Cas9n hiPSCs. The design of the mtagBFP2[30] ssODN donor and the iCRISPR system is shown in **a**. We inserted a 871 nt (including 50 nt homology arms) sequence coding for a 2A-self cleaving peptide in front of a blue fluorescent protein (BFP), directly after the N-terminal nuclear localization signal sequence (NLS) of the Cas9n in the heterozygous AAVS1 iCRISPR locus[13]. If the sequence is inserted, doxycycline will lead to expression of nucleus-imported BFP. Representative images of the mock, NU7026, and CRISPY mix treatment, after 7 days of BFP expression are shown in **b** as phase contrast (PC), propidium iodide nuclei staining (PI), mtagBFP2 expression (BFP), and merge of PI and BFP. Two images (× 50 magnification, white size bar 200 μm) from each of three technical replicates for the respective treatments were used to quantify the percentage of cells with BFP insertion using ImageJ (**c**)

When we used Cas9, which introduces blunt-ended DSBs, no significant effect was seen when adding other small molecules in addition to NU7026 (Fig. 3b). In contrast, the CRISPY mix together with Cpf1 ribonucleoprotein, which produces staggered DNA cuts, introduced by electroporation in 409-B2 hiPSCs, increased TNS 2.9-fold for *HPRT* and 4.0-fold for *DNMT1* (Fig. 3c). Addition of only NU7026 increased TNS 2.1-fold for *HPRT* and 2.4-fold for *DNMT1*. To test whether the CRISPY mix increases TNS in other pluripotent stem cell lines, we edited the gene *KATNA1* in SC102A1 hiPSCs and H9 hESCs using Cas9n plasmid electroporation and *HPRT* in chimpanzee iPSCs using Cpf1 ribonucleoprotein. TNS increased 2.6-fold, 2.8-fold, and 2.3-fold, respectively, and the increase was bigger than when using NU7026 alone (Fig. 3d).

Next, we tested the toxicity of each small-molecule and molecule combinations including the CRISPY mix on iCRISPR 409-B2 hiPSCs using a resazurin assay[29]. After *KATNA1* editing with Cas9n double nicking and CRISPY mix treatment for 24 h cells showed a viability of 75% compared with no small-molecule treatment, with no additive toxic effect of its components (Supplementary Fig. 4A). Importantly, when we simulated five rounds of editing, each round consisting of passaging cells with the lipofection reagent and CRISPY mix followed with 3 days of recovery, the cells had a healthy karyotype with no numerical or large-scale chromosomal aberrations as shown by trypsin-induced Giemsa staining (Supplementary Fig. 4B).

Furthermore, we tested whether the CRISPY mix can also increase efficiency of insertion of a gene fragment. We inserted a 871 nt (including 50 nt homology arms) sequence encoding a 2A-self cleaving peptide in front of an enhanced blue fluorescent protein (BFP)[30] in the AAVS1 iCRISPR locus (Fig. 4a and Supplementary Table 3). If the sequence is inserted, doxycycline will lead to expression of nucleus-imported BFP. Nuclei positive for BFP increased 7.1-fold (26.6%) compared with the no-CRISPY control (3.7%), whereas NU7026 alone lead to an increase of 1.6-fold (6%) (Fig. 4b, c) showing that the CRISPY mix increases efficiency of insertion of a gene fragment in hiPSCs.

**Non-identical small-molecule effects in different cell types.** Finally, we tested whether the CRISPY mix or other combinations of its comprising small molecules increase the efficiency of TNS in non-pluripotent cells. We edited the *HPRT* gene with Cpf1 in two immortalized cell lines (HEK293, K562) and primary cells (CD4$^+$ T cells, CD34$^+$ progenitor cells, and primary human epidermal keratinocytes (HEKa)). TNS percentages are shown in Supplementary Fig. 5 and corresponding cell viabilities after small-molecule treatments are shown in Supplementary Fig. 6. Whereas MLN4924 decreases TNS efficiency in all of those cell lines, other CRISPY components have effects that can differ in different cell lines. NU7026 is the only single small molecule that clearly increases TNS in HEK293 (3.0-fold), K562 cells (4.0-fold),

CD4$^+$ T (3.0-fold), and CD34$^+$ progenitor cells (1.7-fold). However, it decreases TNS in HEKa cells (3.1-fold) (Supplementary Fig. 5). The TNS increase was even higher when the CRISPY mix without MLN4924 was used (6.6-fold and 2.6-fold) in primary CD4$^+$ T cells and CD34$^+$ progenitor cells, respectively (Supplementary Fig. 5B). Admittedly, the achieved TNS efficiencies for CD34$^+$ progenitor cells were very low (increase of 0.24% to 0.63%) and for the other cells lines around 5% with small-molecule treatment, which suggest that the targeted *HPRT* locus is difficult to edit with the donor we used. The treatment with this mix decreased the cell viability to 59 and 65% compared with the electroporation control for CD4$^+$ T cells and CD34$^+$ progenitor cells, respectively (Supplementary Fig. 6B).

## Discussion

Previously, it has been shown that types of cuts introduced by distinct CRISPR enzymes engage different repair pathways, because 5′-overhanging ends yielded higher levels of HDR than 3′-overhangs or blunt ends[7]. This is in line with our observation that Trichostatin A and MLN4924 increase TNS with 5′-overhang-inducing Cas9n and Cpf1 but have no TNS increasing effect with blunt end-inducing Cas9.

In pluripotent stem cells, NU7026, Trichostatin A, MLN4924, and NSC 15520 (CRISPY mix components) increase TNS with Cas9n and Cpf1 when applied either singly or together (Figs 2a and 3a, c and d). NU7026 inhibits DNA-PK (Fig. 1), a major complex in NHEJ pathway[3], and has been previously shown to increase knock-in efficiency in hiPSCs[25]. Trichostatin A activates an ATM-dependent DNA-damage signaling pathway[31]. MLN4924 inhibits the Nedd8-activating enzyme and has been shown to inhibit the neddylation of CtIP, which leads to an increase of the extent of DNA end resection at strand breaks, thereby promoting HDR[32] by leaving ssDNA stabilized by RPA that can undergo recombination. NSC 15520 prevents the association of RPA with p53 and RAD9[33,34], possibly increasing the abundance of RPA available, which could favor HDR. Although RAD51 is obviously important for classical homologous recombination with dsDNA[3], it is possible that RAD52, rather than RAD51, could be responsible for HDR with ssDNA donors, as RAD52 is needed for annealing of ssDNA[6]. Our observation that inhibition of RAD52 by AICAR has no effect on TNS efficiency, while inhibition of RAD51 by B02 halved it, suggests that RAD51 and not RAD52 is important for precise editing with ssODN of both blunt and 5′-staggered ends in hiPSCs. This is in contrast to Bothmer et al.[7] who described that knockdown of RAD51 has no effect on precise editing with ssODN in U2OS cells. RS-1, SCR7, and L755507 for which there are conflicting reports on their capacity to increase precise genome editing (Supplementary Table 1) showed no measurable effect in our hands on TNS neither in the Cas9 or the Cas9n hiPSCs.

Although the CRISPY mix increases TNS more than any individual component it comprised in all four pluripotent stem cell lines tested (three human and one chimpanzee) (Fig. 3a, c, and d), this is not the case for other cell lines tested (Supplementary Fig. 5). In fact, our results (Fig. 3. and Supplementary Fig. 5) show that small molecules and their combinations can have opposite effects on TNS in different cell lines. This could be due to that cell lines rely on different repair proteins or repair pathways.

In line with this interpretation are studies that show that hESCs and iPSCs possess very high DNA repair capacity that decreases after differentiation[35,36]. CtIP expression and proteins levels, and consequently the relative length of resection of DSBs are increased in iPSCs[37], thereby promoting HDR initiation. CtIP

levels can be further artificially increased by inhibiting its neddylation through MLN4924. Our results show that treatment with MLN4924 alone or in a small-molecule mix increases HDR efficiencies when using Cas9n double nicking or Cpf1 in ESCs and iPSCs (Figs 2a, 3a, c, d), whereas it decreases HDR efficiencies in the immortalized and primary cell lines tested (Supplementary Fig 5). Jimeno et al.[32] showed that cell protein neddylation not only affects the choice between NHEJ and HDR, but also controls the balance between different HDR subpathways, as MLN4924 treatment reduced the gene conversion efficiency, whereas it increased single-strand annealing efficiency[32]. It is tempting to speculate that pluripotent stem cells, but not the tested non-pluripotent cells, can efficiently utilize an HDR subpathway that is characterized by CtIP-dependent hyper-resection when confronted with a staggered CRISPR-enzyme-induced DSB and supplied with ssODN. Cell-type-specific reliance on different repair pathways may also explain some of the inconsistencies between studies (Supplementary Table 1), e.g., the DNA ligase IV inhibitor SCR7 and RAD51 enhancer RS-1 increase precise genome editing in some cell types but not in others. Thus, it may be necessary to screen small molecules for their effects on CRISPR editing in each cell type of interest.

In summary, we show that CRISPY mix of small molecules increases TNS in all four pluripotent stem cell lines we tested, after a DSB with 5′-overhangs was introduced with a Cas9n or Cpf1 and a donor DNA was provided as ssODN. We also show that none of the tested small molecules clearly increased TNS in all cell types, which supports the idea of cell-type-specific mechanisms of DNA repair. This suggests that for increasing precise editing efficiency in a cell type of interest the corresponding small-molecule screen needs to be carried out.

## Methods

**Cell culture**. Stem cell lines cultured for this project included human 409-B2 hiPSC (female, Riken BioResource Center) and SC102A1 hiPSC (male, BioCat GmbH), chimpanzee SandraA ciPSC (female, Mora-Bermúdez et al.[38]), as well as H9 hESC (female, WiCell Research Institute, Ethics permit AZ 3.04.02/0118). Stem cell lines were grown on Matrigel Matrix (Corning, 35248) and mTeSR1 (Stem Cell Technologies, 05851) with mTeSR1 supplement (Stem Cell Technologies, 05852) was used as culture media. Non-pluripotent cell types and their respective media used were as follows: HEK293 (ECACC, 85120602) with Dulbecco's modified Eagle's medium/F-12 (Gibco, 31330-038) supplemented with 10% fetal bovine serum (FBS) (SIGMA, F2442) and 1% NEAA (SIGMA, M7145); K562 (ECACC, 89121407) with Iscove's modified Dulbecco's media (ThermoFisher, 12440053) supplemented with 10% FBS; CD4$^+$ T (HemaCare, PB04C-1) with RPMI 1640 (ThermoFisher, 11875-093) supplemented with 10% FBS and activated with Dynabeads Human T-Activator (CD3/CD28) (ThermoFisher, 11131D); CD34$^+$ progenitor (HemaCare, M34C-1) with StemSpan SFEM (Stem Cell, 09600) supplemented with StemSpan CC110 (Stem Cell, 02697); and HEKa (Gibco, C0055C) with Medium 154 (ThermoFisher, M154500) and Human Keratinocyte Growth Supplement (ThermoFisher, S0015). Cells were grown at 37 °C in a humidified incubator gassed with 5% CO$_2$. Media was replaced every day for stem cells and every second day for non-pluripotent cell lines. Cell cultures were maintained 4–6 days until ∼ 80% confluency and subcultured at a 1:6 to 1:10 dilution. Adherent cells were dissociated using EDTA (VWR, 437012 C). The media was supplemented with 10 μM Rho-associated protein kinase (ROCK) inhibitor Y-27632 (Calbiochem, 688000) after cell splitting for one day in order to increase cell survival.

**Generation and validation of iCRISPR cell lines**. 409-B2 hiPSCs were used to create an iCRISPR-Cas9 line as described by Gonzalez et al.[13] (GMO permit AZ 54-8452/26). In brief, the iCRISPR system was introduced using two transcription activator-like effector nucleases targeting the AAVS1 locus and two donors that are responsible for doxycycline-inducible Cas9 expression, namely Puro-Cas9 donor and AAVS1-Neo-M2rtTA. Each inserted cassette has a either a puromycin or a geneticin resistance gene, in order to select for colonies, which have inserted both iCRISPR cassettes. For the production of iCRISPR-Cas9n line Puro-Cas9 donor was subjected to site-directed mutagenesis with the Q5 mutagenesis kit to introduce the D10A mutation (New England Biolabs, E0554S). Primers were ordered from IDT (Coralville, USA) and are shown in Supplementary Table 3. Expression of the pluripotency markers SOX2, OCT-4, TRA1-60, and SSEA4 in iCRISPR lines

was validated using the PSC 4-Marker immunocytochemistry kit (Molecular Probes, A24881) (Supplementary Fig. 7). Quantitative PCR was used to confirm doxycycline-inducible Cas9 or Cas9n expression and digital PCR was used to exclude off-target integration of the iCRISPR cassettes (Supplementary Fig. 8).

**Small molecules**. Commercially available small molecules used in this study were NU7026 (SIGMA, T8552), Trichostatin A (SIGMA, T8552), MLN4924 (Adooq BioScience, A11260), NSC 19630 (Calbiochem, 681647), NSC 15520 (ChemBridge, 6048069), AICAR (SIGMA, A9978), RS-1 (Calbiochem, 553510), Resveratrol (Selleckchem, S1396), SCR7 (XcessBio, M60082-2s), L755507 (TOCRIS, 2197), B02 (SIGMA, SML0364), and STL127685 (Vitas-M). STL127685 is a 4-fluorophenyl analog of the non-commercially available STL127705. Stocks of 15 mM (or 10 mM for NU7026) were made using dimethylsulfoxide (DMSO) (Thermo Scientific, D12345). Solubility is a limiting factor for NU7026 concentration. Suitable working solutions for different concentrations were made so that addition of each small molecule accounts for a final concentration of 0.08% (or 0.2% for NU7026) DMSO in the media. Addition of all small molecules would lead to a final concentration of 0.7% DMSO.

**Design of gRNAs and ssODNs**. We chose to introduce one desired mutation in three genes *CALD1*, *KATNA1*, and *SLITRK1* back to the state of the last common ancestor of human and Neanderthal[39]. gRNA pairs for editing with the Cas9n nickase were selected to cut efficiently at a short distance from the desired mutation and from the respective partnering gRNA. The efficiency was estimated with the sgRNA scorer 1.0 tool[40] as a percentile rank score. Donor ssODNs for nickase editing were designed to have the desired mutation and Cas9-blocking mutations to prevent re-cutting of the locus and had 50 nt homology arms upstream and downstream of each nick (Supplementary Fig. 2). gRNA of the nickase gRNA pair that cuts closer to the desired mutation was used for Cas9 nuclease editing together with a 90 nt ssODN centered at the desired mutation and containing a Cas9-blocking mutation (Supplementary Fig. 2). ssODNs for editing of *HPRT* and *DNMT1* with Cpf1 were designed to contain a blocking mutation near the PAM site and an additional mutation near the cut. gRNAs (crRNA and tracR) and ssODN were ordered from IDT (Coralville, USA). ssODNs and crRNA targets are shown in Supplementary Table 3.

**Lipofection of oligonucleotides**. Cells were incubated with media containing 2 μg/ml doxycycline (Clontech, 631311) 2 days prior to lipofection. Lipofection (reverse transfection) was done using the alt-CRISPR manufacturer's protocol (IDT) with a final concentration of 7.5 nM of each gRNA and 10 nM of the respective ssODN donor. In brief, 0.75 μl RNAiMAX (Invitrogen, 13778075) and the respective oligonucleotides were separately diluted in 25 μl OPTI-MEM (Gibco, 1985-062) each and incubated at room temperature for 5 min. Both dilutions were mixed to yield 50 μl of OPTI-MEM including RNAiMAX, gRNAs and ssODNs. The lipofection mix was incubated for 20–30 min at room temperature. During incubation cells were dissociated using EDTA for 5 min and counted using the Countess Automated Cell Counter (Invitrogen). The lipofection mix, 100 μl containing 25,000 dissociated cells in mTeSR1 supplemented with Y-27632, 2 μg/ml doxycycline and the respective small molecule(s) to be tested were thoroughly mixed and put in 1 well of a 96-well plate covered with Matrigel Matrix (Corning, 35248). Media was exchanged to regular mTeSR1 media after 24 h.

**Ribonucleoprotein electroporation**. The recombinant *A.s.* Cpf1 protein and electroporation enhancer was ordered from IDT (Coralville, USA) and nucleofection was done using the manufacturer's protocol, except for the following alterations. Nucleofection was done using the B-16 program (or U-14 for CD34+ progenitor cells) of the Nucleofector 2b Device (Lonza) in cuvettes for 100 μl Human Stem Cell nucleofection buffer (Lonza, VVPH-5022), or Human T Cell nucleofection buffer for CD4+ T cells (Lonza, VPA-1002), and Human CD34 Cell nucleofection buffer for CD34+ progenitor cells (Lonza, VPA-1003), containing 1 million cells of the respective lines, 78 pmol electroporation enhancer, 0.3 nmol gRNA, 200 pmol ssODN donor (600 pmol for CD4+ T cells), and 252 pmol Cpf1. Cells were counted using the Countess Automated Cell Counter (Invitrogen).

**Fluorescence-associated cell sorting**. Introduction of 2 μg plasmid DNA (pSpCas9n(BB)-2A-GFP (PX461) was a gift from Feng Zhang Addgene 48140[41]) into cells not expressing Cas9 inducably was done using the B-16 program of the Nucleofector 2b Device (Lonza) in cuvettes for 100 μl Human Stem Cell nucleofection buffer (Lonza, VVPH-5022) containing 1 million of either SC102A1 hiPSC or H9 hESC. Cells were counted using the Countess Automated Cell Counter (Invitrogen). Twenty-four hours after nucleofection, cells were dissociated using Accutase (SIGMA, A6964), filtered to obtain a single-cell solution, and subjected to fluorescence-associated cell sorting (FACS) for green fluorescent protein (GFP)-expressing cells. During sorting with the BD FACSAria III (Becton-Dickinson) cells were kept at 4 °C in mTeSR1 supplemented with Y-27632. 48 h after sorting cells were subjected to lipofection with gRNAs, ssODNs, and treatment with small molecules.

**Illumina library preparation and sequencing**. Three days after lipofection cells were dissociated using Accutase (SIGMA, A6964), pelleted, and resuspended in 15 μl QuickExtract (Epicentre, QE0905T). Incubation at 65 °C for 10 min, 68 °C for 5 min, and finally 98 °C for 5 min was performed to yield ssDNA as a PCR template. Primers for each targeted loci containing adapters for Illumina sequencing were ordered from IDT (Coralville, USA) (see Supplementary Table 3). PCR was done in a T100 Thermal Cycler (Bio-Rad) using the KAPA2G Robust PCR Kit (Peqlab, 07-KK5532-03) with supplied buffer B and 3 μl of cell extract in a total volume of 25 μl. The thermal cycling profile of the PCR was: 95 °C 3 min; 34 × (95° 15 s, 65 °C 15 s, 72 °C 15 s); 72 °C 60 s. P5 and P7 Illumina adapters with sample-specific indices were added in a second PCR reaction[42] using Phusion HF MasterMix (Thermo Scientific, F-531L) and 0.3 μl of the first PCR product. The thermal cycling profile of the PCR was: 98 °C 30 s; 25 × (98° 10 s, 58 °C 10 s, 72 °C 20 s); 72 °C 5 min. Amplifications were verified by size separating agarose gel electrophoresis using EX gels (Invitrogen, G4010-11). The indexed amplicons were purified using Solid Phase Reversible Immobilization (SPRI) beads[43]. Double-indexed libraries were sequenced on a MiSeq (Illumina) giving paired-end sequences of 2 × 150 bp. After base calling using Bustard (Illumina), adapters were trimmed using leeHom[44].

**CRISPResso analysis**. CRISPResso[45] was used to analyze sequencing data from CRISPR genome-editing experiments for percentage of wild type, TNS, indels, and mix of TNS and indels. Parameters used for analysis were "-w 20," "–min_identity_score 70," and "–ignore_substitutions" (analysis was restricted to amplicons with a minimum of 70% similarity to the wild type sequence and to a window of 20 bp from each gRNA; substitutions were ignored, as sequencing errors would be falsely characterized as NHEJ events).

**Statistical analysis**. Significances of changes in TNS efficiencies were determined using a two-way analysis of variance and Tukey's multiple comparison pooled across the three genes *CALD1*, *KATNA1*, and *SLITRK1*. Genes and treatments were treated as random and fixed effect, respectively. Hence, we tested the effect of treatment against its interaction with gene[46]. Analysis included three technical replicates for each gene. We checked for whether the assumptions of normally distributed and homogeneous residuals were fulfilled by visual inspection of a QQ-plot[47] and residuals plotted against fitted values[48]. These indicated residuals to be roughly symmetrically distributed but with elongated tails (i.e., too large positive and negative residuals) and no obvious deviations from the homogeneity assumption. *P*-values are adjusted for multiple comparison. Statistical analysis was done using R.

**Resazurin assay**. 409-B2 iCRISPR-Cas9n hiPSCs were either seeded with or without editing reagents (RNAiMax, gRNA, and ssODN donor for *KATNA1* editing) as described in "Lipofection of oligonucleotides" (25,000 cells per 96 wells). Non-pluripotent cell lines were either seeded without editing reagents or electroporated with editing reagents as decribed in "Ribonucleoprotein electroporation" (50,000 cells per 96 wells). The media was supplemented with small molecules or combinations of small molecules, and each condition was carried out in duplicate. After 24 h, media was aspirated and 100 μl fresh media together with 10 μl resazurin solution (Cell Signaling, 11884) was added. Resazurin is converted into fluorescent resorfin by cellular dehydrogenases and resulting fluorescence (exitation: 530–570 nm, emission: 590–620 nm) is considered as a linear marker for cell viability[29]. Cells were incubated with resazurin at 37 °C. The redox reaction was measured every hour by absorbance readings using a Typhoon 9410 imager (Amershamn Biosciences). After 5 h (12 h for CD34 + progenitor cells) the absorbance scan showed a good contrast without being saturated, and was used to quantify the absorbance using ImageJ and the "ReadPlate" plugin. Duplicate wells with media and resazurin, but without cells, were used a blank.

**Microscopy and image analysis**. 409-B2 iCRISPR-Cas9n hiPSCs were electroporated with gRNAs and the BFP single-stranded oligo (Fig. 4a, 330 ng) in two technical replicates for either mock, NU7026, and CRISPY mix treatment. Media was supplemented with 2 μg/ml doxycycline (Clontech, 631311) for 7 days, to allow expression of nuclear imported BFP in precisely edited cells. Then, cells were fixed with 4% formaldehyde in Dulbecco's phosphate-buffered saline (DPBS) (ThermoFisher, A24881) for 15 min, permeablized with 1% saponin in DPBS (ThermoFisher, A24881) supplemented with 100 μg/ml RNAseA (ThermoFisher, EN0531) and 40 μg/ml propidium iodide (ThermoFisher, P3566) for 45 min at 37 °C, and washed three times with DPBS. Nucleic acid intercalating propidium iodide was used to counterstain nuclei. A fluorescent microscope Axio Observer Z (Zeiss) was used to aquire two images (× 50 magnification), from each of three technical replicates for the respective treatments, consisting of the following: phase contrast, HcRed channel (BP 580–604 nm, BS 615 nm, BP 625–725 nm, 10.000 ms), and 4',6-diamidino-2-phenylindole channel (BP 335–383 nm, BS 395 nm, BP 420–470 nm, 20,000 ms). Images were blinded and BFP-positive nuclei were counted using the Adobe Photoshop CS5 counting tool. Propidium iodide-positive nuclei were quantified using ImageJ by dividing the area of nuclei (default threshold) with the mean area of a single nuclei.

**Karyotyping**. Microscopic analysis of the karyotype was done after trypsin-induced Giemsa staining. The analysis was carried out according to international quality guidelines (ISCN 2016: An International System for Human Cytogenetic Nomenclature[49]) by the "Sächsischer Inkubator für klinische Translation" (Leipzig, Germany).

**Data availability**. Data available on request from the authors.

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

## Acknowledgements

We thank Malgorzata Santel for assistance with FACS sorting, Anna Kirstein for experimental help in validation of the iCRISPR cell lines, Roger Mundry for help with statistical analysis, and Heidrun Holland for karyotyping. Furthermore, we thank Antje Weimann and Barbara Schellbach for DNA sequencing, and Adrian Briggs and especially Svante Pääbo for comments on the manuscript and helpful discussions. This work was supported by the Max Planck Society and by the NOMIS Foundation.

## Author contribution

S.R. conceived the idea. S.R. and T.M planned the experiments. S.R. performed the experiments. S.R. and T.M. wrote the paper.

## Additional information

**Competing interests:** A related patent application on compounds for increasing genome-editing efficiency has been filed (patent applicant: Max Planck Society, inventors: S.R. and T.M., application number: EP17203591.7, PCT/EP2018/059173, status: pending).

