## [Peer Review File · Nature Communications]

Reviewers' comments:

Reviewer #2 (Remarks to the Author):

The authors have responded in part to our initial critique, but not completely. The revised manuscript still raises the following concerns.

(1) Toxicity is still a problem for this study. The revised manuscript includes evaluation of toxicity in iPSCs (for a subset of experiments), but is not included in, for example, the experiments in primary CD4+ or CD34+ cells, which we feel is a major oversight that limits the usefulness of this data.

(2) We find the writing still to be unclear in places, very "list-like" and with missing information - for example, figuring out what gene, cell line, and HR assay is being used is difficult to find in either the text, figure legends, or figures themselves.

(3) The primary cell data in Supp Fig 5B is of concern because (i) the editing levels reported are extremely low compared to what is now standard to achieve in these cells, (ii) the error bars are huge, (iii) replicates only comprise two technical replicates of two independent experiments, and (iv) the data lacks appropriate accompanying analyses such as toxicity. The data shows that, at best, a version of the drug mix increases gene editing 3.6-fold (CD4+ cells) and 2.6 fold (CD34+ cells), with the gene editing rates going from 0.75% to 1.4% (CD4+) and 0.5% to 1.75% (CD34+). I therefore find that their conclusions are not justified and recommend that these analyses either be performed more rigorously, or not included in the manuscript.

(4) The authors need should make it clear throughout the paper (in their title and the main text) that their conclusions and evidence of efficacy are mostly based on working with a pluripotent iPSC cell line that expresses Cas9, in combination with ssODNs, and are not the broader applications that the title implies.

(5) The impact of the overall study is reduced by the fact that although some HDR increases were achieved in engineered (Cas9-expressing) iPSCs, the mix doesn't work in other cell types, as stated above, there is no compelling evidence that it works in primary cells, and there is no proposed explanation or mechanism to guide the use of the drug mix in other cell types. Indeed, the authors acknowledge that such experiments would need to be done empirically.

Reviewer #3 (Remarks to the Author):

The authors have properly addressed all my concerns. The manuscript is ready to publish with only one change: they should consider move Figure 1 to supplemental. Including such a conceptual figure as Figure 1 is a bit distracting to the whole story.

Reviewers' comments:

Reviewer #2:

The authors have responded in part to our initial critique, but not completely. The revised manuscript still raises the following concerns.

(1) Toxicity is still a problem for this study. The revised manuscript includes evaluation of toxicity in iPSCs (for a subset of experiments), but is not included in, for example, the experiments in primary CD4+ or CD34+ cells, which we feel is a major oversight that limits the usefulness of this data.

We performed experiments and added new data as requested (page 6 end of results section, Supplementary Fig. 6)

(2) We find the writing still to be unclear in places, very list-like; and with missing information - for example, figuring out what gene, cell line, and HR assay is being used is difficult to find in either the text, figure legends, or figures themselves.

We think that it is an inherent characteristic of our study which systematically tests many small molecules to have lists in our text (e.g. small molecules and proteins they interact with). In order to help the reader we put effort into making Figure 1, Supplementary Table 1, 2 and 4 so that it is easier to read through some lists in the text.

We checked now all the figures and found the information on the genes, cell lines and HR assay existed in all of them but in Figure 4 which we now changed to include the missing information.

(3) The primary cell data in Supp Fig 5B is of concern because (i) the editing levels reported are extremely low compared to what is now standard to achieve in these cells, (ii) the error bars are huge, (iii) replicates only comprise two technical replicates of two independent experiments, and (iv) the data lacks appropriate accompanying analyses such as toxicity. The data shows that, at best, a version of the drug mix increases gene editing 3.6-fold (CD4+ cells) and 2.6 fold (CD34+ cells), with the gene editing rates going from 0.75% to 1.4% (CD4+) and 0.5% to 1.75% (CD34+). I therefore find that their conclusions are not justified and recommend that these analyses either be performed more rigorously, or not included in the manuscript.

We have now done additional experiments to address the concerns (Supplementary Fig. 5B and 6). We tried to increase the TNS efficiencies by supplying 600pmol instead of 200pmol ssODN. We were successful in increasing the TNS efficiency in CD4+ T cells to almost 5% in 3 independent experiments (levels comparable in HEK293, K562, and HEKa cells when using Cpf1 to target *HPRT*) (i, iii). The error bars are now smaller for CD4+ T cells (ii). When supplying 600pmol ssODN to CD34+ progenitor cells the overall editing efficiency dropped, thereby further reducing TNS efficiency rates. Since the TNS efficiency is very low for CD34+ progenitor cells the error bars appear naturally bigger (ii). We doubled the replicates for CD34+ progenitor cells (now four replicates, 200pmol ssODN) and now we state in the text (page 5, line 10-11) that the efficiencies are very low for CD34+ cells (i,iii). We added toxicity data in Supplementary Fig. 6 (iv).

(4) The authors need should make it clear throughout the paper (in their title and the main text) that their conclusions and evidence of efficacy are mostly based on working with a pluripotent iPSC cell line that expresses Cas9, in combination with ssODNs, and are not the broader applications that the title implies.

As we show in Figure 3 already, the small molecule mix increases TNS efficiency regardless if we used Cas9 nickase or Cpf1 delivered either in a plasmid (Fig. 3D) or protein form (Fig. 3C and D) or if Cas9n

expression is induced in the cell line (3A). This shows that the mix increases TNS regardless of the way the CRISPR enzyme is delivered.

It is now common practice in genome editing that ssODNs are used for introduction of both substitutions and gene-fragment insertions up to 2kb (e.g. in our Figure 4). We already mention using ssODN donors in the main text (page 3, line 10; page 4, line 19), in the methods (page 9 'Design of gRNAs and ssODNs'), give the list of all the ssODNs in Supplementary Table 3 and finally we now include a sentence in a summarizing paragraph at the end of the introduction (page 3 last paragraph) the summary of the paper (page 7 end of discussion).

(5) The impact of the overall study is reduced by the fact that although some HDR increases were achieved in engineered (Cas9-expressing) iPSCs, the mix doesn't work in other cell types, as stated above, there is no compelling evidence that it works in primary cells, and there is no proposed explanation or mechanism to guide the use of the drug mix in other cell types. Indeed, the authors acknowledge that such experiments would need to be done empirically.

We show that the CRISPY mix increases editing in all four pluripotent stem cell lines we tested, regardless of the delivery of the CRISPR enzymes as long as 5' overhangs are produced (Figure 3). That is why the focus of the paper is genome editing in pluripotent stem cells, which is reflected even in the title. We wish we could provide one mix for all cell types but the reality is that different cell types need different mixes which suggests that repair pathways are specific for a cell type; we discuss this more now at page 6 and 7.

Reviewer #3 (Remarks to the Author):

The authors have properly addressed all my concerns. The manuscript is ready to publish with only one change: they should consider move Figure 1 to supplemental. Including such a conceptual figure as Figure 1 is a bit distracting to the whole story.

Figure 1 is an illustration of relevant repair proteins and small molecules that target them directly or indirectly. We considered moving the figure to the supplementary material but have decided against it since we find it useful for a reader to follow the main text and the design of the study if Figure 1 is easily accessible.

REVIEWERS' COMMENTS:

Reviewer #1 (Remarks to the Author):

The authors have addressed my most pressing concerns from the December 2017 review. They have adequately analyzed and presented toxicity of their treatments on their various cell types, as well as clarified the method of presentation. Their additional supplemental tables describing their results are helpful.

I continue to note their exceptionally low levels of editing in CD4+ T cells, CD34+ progenitor cells, and primary HEKa cells, which are still concerning. They argue that low editing levels are acceptable due to them paralleling their editing levels in K562s, which does not seem like an appropriate argument, since their K562 TNS editing levels are also exceptionally low and these are usually considered easy cells to edit.

Reviewer #1 comments:

The authors have addressed my most pressing concerns from the December 2017 review. They have adequately analyzed and presented toxicity of their treatments on their various cell types, as well as clarified the method of presentation. Their additional supplemental tables describing their results are helpful.

I continue to note their exceptionally low levels of editing in CD4+ T cells, CD34+ progenitor cells, and primary HEKa cells, which are still concerning. They argue that low editing levels are acceptable due to them paralleling their editing levels in K562s, which does not seem like an appropriate argument, since their K562 TNS editing levels are also exceptionally low and these are usually considered easy cells to edit.

We agree with the reviewer that the efficiency of precise editing is low, but the reality is that many loci do not have high precise editing efficiencies and especially for those loci it is important to increase the efficiency. For the locus in question (*HPRT*) we increase the efficiency to around 5% with small molecule treatment, which is an increase that can be easily detected by NGS sequencing.

We included an additional sentence at the end of the results: "Admittedly, the achieved TNS efficiencies for CD34+ progenitor cells were very low (increase of 0.24% to 0.63%) and for the other cell lines around 5% with small molecule treatment, which suggests that the targeted *HPRT* locus is difficult to edit with the donor we used."